# Experimental Methods for the Biological Evaluation of Nanoparticle-Based Drug Delivery Risks

**DOI:** 10.3390/pharmaceutics15020612

**Published:** 2023-02-11

**Authors:** Ramendra Pati Pandey, Jasmina Vidic, Riya Mukherjee, Chung-Ming Chang

**Affiliations:** 1Centre for Drug Design Discovery and Development (C4D), SRM University, Delhi-NCR, Rajiv Gandhi Education City, Sonepat 131 029, Haryana, India; 2AgroParisTech, The Institut National de la Recherche Agronomique (INRAE), Micalis Institute, Université Paris-Saclay, 78350 Jouy-en-Josas, France; 3Graduate Institute of Biomedical Sciences, Division of Biotechnology, Chang Gung University, No. 259, Wenhua 1st Rd., Guishan District, Taoyuan City 33302, Taiwan; 4Master & Ph.D. Program in Biotechnology Industry, Chang Gung University, No. 259, Wenhua 1st Rd., Guishan District, Taoyuan City 33302, Taiwan

**Keywords:** biological models, liposomes, nanoparticles, drug delivery, advanced technologies, in vitro/in vivo correlation

## Abstract

Many novel medical therapies use nanoparticle-based drug delivery systems, including nanomaterials through drug delivery systems, diagnostics, or physiologically active medicinal products. The approval of nanoparticles with advanced therapeutic and diagnostic potentials for applications in medication and immunization depends strongly on their synthesizing procedure, efficiency of functionalization, and biological safety and biocompatibility. Nanoparticle biodistribution, absorption, bioavailability, passage across biological barriers, and biodistribution are frequently assessed using bespoke and biological models. These methods largely rely on in vitro cell-based evaluations that cannot predict the complexity involved in preclinical and clinical studies. Therefore, assessing the nanoparticle risk has to involve pharmacokinetics, organ toxicity, and drug interactions manifested at multiple cellular levels. At the same time, there is a need for novel approaches to examine nanoparticle safety risks due to increased constraints on animal exploitation and the demand for high-throughput testing. We focus here on biological evaluation methodologies that provide access to nanoparticle interactions with the organism (positive or negative via toxicity). This work aimed to provide a perception regarding the risks associated with the utilization of nanoparticle-based formulations with a particular focus on assays applied to assess the cytotoxicity of nanomaterials.

## 1. Introduction

Nanotechnology is an interdisciplinary branch that studies particles sized between 1 and 100 nanometers at least at one dimension. Nano-sized particles exhibit a high surface area-to-volume size ratio and show unique properties that enable a variety of applications in medical, engineering, agri-food, and related sectors. Moreover, the interactions of nanoparticles (NPs) with biomolecules and their behavior in cell, tissue and organism contexts has allowed the development of nanoparticle-based therapy approaches [1]. Nanomedicine aims to overcome the limits of free therapies and biological hurdles that vary across patient groups and diseases. Pharmaceutical dosage formulations based on NPs and different drugs (such as cytostatics, proteins, peptides, ARN, antibiotics, and antiviral and antiparasitic drugs) have become a high priority in pharmaceutical research. The use of lipid nanoparticles as a carrier for siRNA and mRNA in a recently approved medication and vaccine illustrates advances in the field. Novel engineered nanomaterials still hold much promise for improving disease diagnosis and specific treatment [2]. The principal advantage of NP-drug complexes is their ability to reach tissue and target organs while improving the drug’s intracellular penetration and distribution. In addition, nanodevices for drug delivery may protect a drug from degradation and allow the modification of drug pharmacokinetics.

Two main approaches regarding nanoparticle-based drug delivery are under development: the optimization of delivery systems with a one-size-fits-all approach and the precise engineering of nanoparticles (lipid-based, polymeric, and inorganic) for more individualized medication administration, ushering in the time of targeted therapy [3]. Precision treatments, in which individualized approaches increase therapeutic success, also help to overcome patient heterogeneity [4]. Other medical applications of advanced nanomaterials include tissue regeneration therapy, implants, imaging contrast agents and nanostructured devices or devices that contain nanoparticles [5,6]. Since pathophysiology will continue to grow as the global population ages and the world advances, putting a substantial physical and financial impact on total social insurance systems, the development of bioactive NPs is expected to grow tremendously in the future [7].

Nanoparticles and nano-formulations may act differently from their bulk molecules and substances of the same composition. In particular, due to their small size and high surface area, NPs can be significantly more effective than conventional materials of the same composition [8]. Despite the pivotal clinical advantages, highly active NPs are likely to have negative effects. There are several approved orally given nano-formulations that may cause severe secondary effects due to their topical toxicities to the gastrointestinal system and metabolic organs, such as the liver and kidney. Due to their small size and accumulated surface charge, surface tension, and high chemical/structural complexity, nanoparticles may penetrate different organs and cell compartments [9]. Typically, nanoparticles are taken up through endocytosis by the cells in the liver, spleen, lungs and bone marrow [10]. Consequently, it is important to elucidate the fate of internalized NPs and immune responses to them because both can differ from those elicited by standard formulations containing particles of larger sizes.

Engineered nanoparticles with pharmacological potential can be rapidly taken up by a variety of cell types and have the potential to traverse intracellular and intercellular barriers [11]. NPs can trigger the production of reactive oxygen species, activate the complement system, or impair the functionality of membranes and cellular barriers, depending on the kind, dose, and incubation period. These acts cause immediate or persistent damage to the organism, which can result in catastrophic consequences such as inflammation, gene mutations, and severe organ damage [11]. Therefore, the benefit-to-risk ratio has to be estimated regarding the intended medical application.

In this paper, we discuss the existing applications of nanoparticle-based formulation and the risks associated with the utilization of nanomaterials, with a particular focus on biological evaluation. Additionally, a strong emphasis is given to assays mostly applied to assess the possible cytotoxic nature of novel nanoparticle-based formulations.

## 2. Drug Delivery Applications

Over the past few years, studies of nano-sized drug delivery systems have become a flourishing research field, and many formulations have reached the market [12]. However, the administration of some NPs is associated with an increased risk of toxicity, requiring discontinuation of the therapy. Clinical pharmacologists make efforts to produce safe nanomedicines by combining engineered nanoparticles with precise control over their surface modifications (such as surface charge, covertness, size, shape, and targeting moieties) and other characteristics that can be screened in order to find the best formulation assuring a prolonged and tailored release with low toxicity. Moreover, the tendency is to make drug delivery systems multifunctional and programmable by external signals or the local environment, thereby transforming them into nanodevices.

The crucial requirement of efficiently established technology is to precisely deliver drugs to diseased areas in the body together with tissue biodistribution and rapid metabolization and excretion from the body. Several methods to obtain drug delivery systems have been developed: (i) drug physical encapsulation into biocompatible nanoparticle assemblies during the formulation process, (ii) the self-assembly of polymers in an aqueous solution containing the drug, (iii) growing a single polymer chain from a solution containing a drug in a controlled fashion, termed “drug-initiated”, and (iv) drug conjugation with a nano-promoiety (Figure 1). Usually, the drug is inactive when conjugated but active when the nanocarrier is cleaved. This method is frequently used for liposome-based and polymer-based formulations, in which drugs are covalently bound to lipidic or polymer scaffold building blocks. The cleavage may be induced by hydrolysis, enzymatic reactions, or reduction, and the active drug is realized from the polymer. However, although simple synthetic methods for producing nanoparticle-based emulsions are highly desired, the development of nanocarriers often requires a series of synthetic steps to ensure stability and protection and decrease toxicity. An emerging approach is based on paramagnetic nanoparticles enabling the remote directing and management of the drug delivery operations, such as driving magnetic nanoparticles to the tumor and then either releasing the drug load or just heating them to destroy the surrounding tissue [13,14,15].

Examples of nanocarriers that have gained traction in the pharmaceutical industry are liposomes or lipid-based nanoparticles. Lipid nanoparticles, in particular, have undergone extensive research and have effectively accessed the clinical field for the delivery of small molecules, siRNA medicines, and messenger RNA (mRNA) [18,19]. Small interfering RNAs (siRNAs) are used to mediate gene silencing in cells, and RNA interference (RNAi) is an emerging cancer therapeutic method [20]. To obtain the efficient distribution of siRNAs into cells in vivo, including tumor and/or host cells in the tumor microenvironment, successful RNAi-mediated gene silencing requires overcoming numerous physiological barriers. Lipid-based nanoparticle siRNA delivery techniques allow for overcoming these physiological hurdles. Because of their significant negative charge, siRNAs can be integrated into NP formulations via covalent connections with lipid components or electrostatic interactions with the liposome surface [21]. Another example is the encapsulation of mRNA, which is a new type of therapeutic agent that can be used to prevent and treat a variety of ailments. To be functional in vivo, mRNA requires delivery mechanisms that are safe, effective, and stable, as well as systems that allow for cellular uptake and mRNA release [19]. Lipid nanoparticle–mRNA vaccines have started to be in clinical use against coronavirus disease 2019 (COVID-19), which really is a major step forward for mRNA therapies [22]. Protein replacement therapies, viral vaccines, cancer immunotherapies, cellular reprogramming, and genome editing are just a few of the instances where mRNA has demonstrated therapeutic potential, attracting the interest of formulation scientists due to its low toxicity, excellent drug solubility, substance release, and precise targeting.

## 3. Biological Barriers in Drug Delivery Therapy

The delivery of medications to a target place typically entails traversing biological barriers. Biological barriers keep organs and tissues safe from physical, chemical, and biological injury while also maintaining tissue homeostasis [23]. The biological barriers also serve as key interfaces among organs and their exterior, such as body fluids. Endothelial or epithelial cell layers are the fundamental components of biological barriers [24]. The barriers are semi-permeable since they keep extraneous material out of the tissue while allowing small molecules of specific characteristics to pass through. Consequently, tissue-specific nanocarriers that can cross the biological barrier due to their small sizes, morphology, and surface chemistry can transport bigger molecules [23]. Biological barriers could be a direct target for treatment techniques [25] when NPs are used to disturb and weaken the barrier in order to increase its permeability. The paracellular barrier, efflux of molecules, the metabolic barrier, signaling between the body fluid and the tissue, and waste clearance from the tissues are all affected by these functional changes. The features of biological barriers constitute both a difficulty in drug delivery and an opportunity for developing custom medication delivery systems that effectively reach the target region. The dynamics of NP fluid in blood vessels depend on particles’ size, surface charge, rigidity, and structural topography. By designing the physicochemical properties of an NP, its biodistribution and half-life can be improved. Most formulations have been developed for intravenous drug administration. The surface charge has been shown to crucially affect the pharmacokinetics and biodistribution of nanoparticles in blood vessels through its role in protein adsorption. Highly positively charged nanoparticles are usually more rapidly cleared from circulation than highly negatively charged nanoparticles. In contrast, neutral, slightly positive and slightly negative NPs may circulate in blood with prolonged half-lives [26].

The functionality of biological barriers is known to be affected by various disorders. It is worth noting that biological barriers are dynamic systems and that small perturbations in their microenvironments may induce changes. The biological barriers, such as crossing epithelial barriers, intracellular delivery, navigating tumor micro-environments, and targeting immune cells, can be overcome by the nanoparticles used as the carrier to reach the targeted site [27]. Modifications in dynamic barrier properties are sometimes difficult to predict. This may represent a challenge for drug delivery development because the irreversible manipulation of biological systems can cause severe side effects [28]. However, this specific dynamic also presents an opportunity for the nanocarrier-based manipulation of biological barriers and facilitation of drug delivery.

## 4. Nanoparticle Toxicity

The physical and chemical features of NPs, such as their size, shape, surface charge, chemical compositions of the core and shell, morphology, and stability, can influence their toxicity. Among them, size and shape seem to be crucial factors that influence the particles’ interaction with living systems. Understanding the mechanism of nanoparticle toxicity provides a basis to redesign them and reduce the side effects of nano-formulations. The redesign has to take into account both the decline of the major mode of toxicity and the need to preserve the nanomaterial’s ability to perform its activity in its intended application.

### 4.1. Nanoparticle Size, Surface Area and Toxicity

The size of the NPs has a significant impact on their interactions with the transport and defense systems of cells and the body and thus plays a central role in determining particle activity in biomedical applications. In the case of inorganic NPs, their chemical properties and solubility are size-dependent. Colloidal solutions can be prepared with particles with diameters up to 100–200 nm under conditions where larger nanoparticles of the same material usually precipitate. As mentioned above, to assess certain biological barriers and compartments, small sizes are needed. The size restrictions may come from steric effects or specific biological functions [29]. Although decreasing the nanocarrier’s size offers many advantages, it can also enhance its toxicity. The in vivo evaluation of subcellular location, tissue distribution, and toxicity of gold NPs in rats has shown that small particles (10 and 30 nm) crossed the cell membrane and membrane of the nucleus and damaged DNA, but particles of 60 nm did not have this effect [30]. In addition, it was shown that gold NPs of 10 and 30 nm highly accumulated in the liver, kidney, and intestine, while the highest accumulation of 60 nm gold NPs was observed in the spleen.

The increased specific surface area ensures that NPs adhere efficiently to the cell and tissue surfaces. Particles smaller than 100 nm were efficiently adsorbed on the erythrocyte surface without causing cell death or morphological abnormalities, whereas particles larger than 600 nm distorted the membrane and entered the cells, causing erythrocyte death [31]. In some cases, engineered NPs with a high surface area and reactivity can generate a high level of reactive oxygen species, even intracellular ROS, thus leading to cytotoxicity and genotoxicity [32].

### 4.2. Nanoparticle Shape and Toxicity

Nanomaterials can be classified regarding their shape as 0D (spherical particles such as carbon and quantum dots or nanoparticles), 1D (materials with one dimension < 100 nm, such as nanowires, nanotubes, and nanorods), 2D (materials with two dimensions < 100 nm, such as nanodisks and nanosheets) and 3D (material with three dimensions < 100 nm, such as nanoflowers, nanoballs, and nanocones) [6]. Among nanomaterial spheres, ellipsoids, cylinders, sheets, cubes, and rods are the most common shapes. Their toxicity is highly influenced by their form. For instance, when the effect of needle-like, plate-like, rod-like, and spherical hydroxyapatite NPs was tested on grown BEAS-2B cells, it was found that plate-like and needle-like NPs killed a higher percentage of cells than spherical and rod-like NPs [33].

### 4.3. Nanoparticle Chemical Composition and Toxicity

Although the size and shape of NPs have a substantial influence on their toxicity, other aspects, such as the NP’s chemical composition and crystal structure, should not be overlooked. It has been demonstrated that NPs can degrade, and the extent of this degradation is dependent on environmental factors such as pH and ionic strength [32]. The most prevalent cause of NPs interacting with cells becoming hazardous is metal ion leakage from the NP core. Toxicity is also affected by the NPs’ core makeup. Some metal ions, such as Ag^+^ and Cd^2+^, are intrinsically poisonous, and their liberation induces cell damage. Other metal ions, such as Fe^3+/4+^, Mg^2+^, and Zn^2+^, are essential oligo-elements and therapeutically helpful, but in a high amount, can impair cellular processes, resulting in significant toxicity [34]. This effect can be reduced by replacing toxic species with less toxic substances that have similar properties or wrapping NP cores with robust polymer shells, silica layers, or gold shells instead of weak ligands. The chemical stabilization of the nanomaterials may prevent degradation and metal ion leakage into the body. The constitution of the core, on the other hand, might be changed by doping with different metals. For instance, the utilization of TiO_2_ has raised concerns about its toxicity, and the European Union banned its use at the beginning of 2022. Recent research focused on iron titanate (Fe_2_TiO_5_) nanoparticles presented as a possible biocompatible alternative to TiO_2_ [35]. Fe_2_TiO_5_ NPs of an average particle size of 44 nm and rhombohedral morphology caused no cell damage to human Caco-2 epithelial cells, as demonstrated by acridine orange cell staining followed by flow cytometry analysis. Alternatively, a chelating agent can be administrated together with the active nanomaterial or functionalized onto its surface to prevent toxic metal migration into the body. Finally, the morphology of the nanoparticle can be designed to minimize surface area and thus minimize dissolution [36].

### 4.4. Nanoparticle Surface Charge and Toxicity

Because the interactions of NPs with biological systems are largely determined by their surface charge, the surface charge of NPs plays an important role in their toxicity. The capacity of positively charged NPs to easily enter cells, as opposed to negatively charged and neutral NPs, explains their greater toxicity [37]. Positively charged NPs have a greater ability to opsonize or adsorb proteins that aid phagocytosis, such as antibodies and complement components, from blood and biological fluids. To modify the surface charge of nanomaterials, they can be produced by synthetic routes that generate a negative surface charge or can be designed to carry ligands such as polyethylene glycol that reduce protein binding. In turn, such modifications in surface properties may discourage particles from binding to cell surfaces and allow them to be controlled in terms of localization, which is highly important in the development of effective methods for delivering therapeutic medications to targets.

To reduce the production of reactive oxygen species, the band gap of the material can be tuned either by using different elements or by doping, a shell layer can be added to inhibit direct contact with the core, or antioxidant molecules can be tethered to the nanoparticle surface. When redesigning nanoparticles, it will be important to test that the redesign strategy actually reduces toxicity to organisms from the relevant environmental compartments. It is also necessary to confirm that the nanomaterial still demonstrates the critical physicochemical properties that inspired its inclusion in a product or device.

The use of microorganisms or plant extracts to synthesize nanoparticles allows for obtaining nanoparticles with high biocompatibility. Recent research showed that nanoparticles synthesized by such green synthesis methods have surfaces coated with proteins, fibers, and carbohydrates that provide them greater biocompatibility than those of the same size and shape but synthesized using chemical methods [38,39].

## 5. Biological Evaluation of Nanoparticle-Based Formulations

Biodistribution, metabolic destiny, non-degradable system resistance, specific therapeutic difficulties, and immunogenicity are all considered in the biomedical evaluation of nanomedicines. Thus, biological evaluation of nanoparticle-based formulations requires the intensive toxicological research [40]. It is vital to determine the full range of hazardous consequences that any nanomaterial may have when used intentionally or inadvertently. According to previous research, inflammatory stimuli, inflammatory cytokine overproduction, increased reactive oxygen and nitrogen species production (RONS) are associated with the majority of nanomaterial-induced initial toxic effects, enroute to any of the apoptosis, necrosis, or autophagy-mediated cell death mechanisms, ultimately leading to cytotoxicity [41,42].

Despite having the same size and chemical composition as their bulk biopharmaceuticals, NPs can cause unexpected toxicity due to their high surface-to-volume area, which increases reactivity, generates band gap modifications, lowers their melting point, and creates major adverse effects [15]. The routine in vitro evaluation of the toxicity and genotoxicity of nanoparticles is performed before subjecting them to any test involving animals in order to minimize the utility of the animals. Table 1 shows the main conventional methods performed to assess the toxicity of nano-based formulations. In many studies, several in vitro tests have been run to measure the toxicity of the nanoparticles because conventional tests may fail to provide results in accordance with effects observed in vivo [43]. However, in vitro tests are useful for determining initial toxicity, whereas in vivo models can provide information on subsequent consequences, including inflammation [43].

In addition to cytotoxicity and genotoxicity, nanomaterials may induce inherited genetic changes in genetic expression that emerge without changes in DNA sequence, referred to as epigenetics. The interplay of three basic mechanisms—DNA methylation, histone modifications, and RNA-mediated post-transcriptional regulation—determines the epigenetic realm. Because of their pro-oxidative qualities, different nanomaterials can indirectly alter DNA methylation [42]. DNA methyltransferases can be hampered by oxidatively damaged DNA. These changes have the potential to affect DNA methylation and histone modification patterns on a wide scale. However, there is still a scarcity of data on nanomaterial-induced histone protein changes. In terms of the dysregulation of microRNA (miRNA) expression profiles, several nanomaterials have shown epigenetic effects [44].

Immunotoxicity can be induced by nanomaterials interacting with immune-competent cells. Nanomaterials may trigger apoptosis and necrosis in immune cells, and their interactions with the immune response can alter immune-specific signaling pathways, culminating in alterations in immune cell function as evaluated by surface marker expression, cytokine generation, cell differentiation, and immunological activation [45]. Furthermore, autoimmune reactions can be triggered by host-protein interactions with nanomaterials and their persistence in the body [46].

Although a large amount of nanomedicine is dedicated to fighting cancer, the risk of nanomaterials producing cancer is equally considerable [47]. Because of their small surface-to-volume area and size, the carcinogenic potential of nanomaterials is thought to be larger than that of conventional materials. Nanomaterials that may cause cancer should be identified, and exposure to them should be limited. In vivo carcinogenicity tests in laboratory animals are the “gold standard” method to evaluate the carcinogenic potential of NPs. However, because the in vivo tests make use of a high number of animals, are time-consuming, expensive, and need ethical approval by the authorities, different in vitro cutting-edge technologies known as cell transformation assays have been developed. For instance, the cell transformation assay cab offers a revolutionary approach that can predict cells’ potential to convert to cancer cells in a single step [48]. Typically, carcinogenicity studies in vitro are performed to complement in vitro genotoxicity test batteries to identify non-genotoxic carcinogens. However, to reduce the risk of exposure to carcinogenic nanomaterials that are constantly under development, the standardization of testing methodologies is still required.

## 6. Experimental Models for Evaluating Nanoparticle-Based Formulations

Customized protocols or experimental models to evaluate the efficacy of targeting and associated risks include distribution, clearance, haematology, serum chemistry, and histopathology [49]. Figure 2 illustrates the complexity of some of these models. Biodistribution studies look at how nanoparticles find their way into a tissue or organ. Radiolabels can be used to trace nanoparticles in dead or alive animals. The measurement of nanoparticle excretion and metabolism at various time periods following exposure is used to determine their clearance [50]. Examining changes in serum chemistry and cell type following nanoparticle exposure is another way to determine in vivo toxicity. The cytotoxicity level induced by a nanoparticle is determined by the histopathology of the cell, tissue, or organ following exposure [51]. Below, the conventional methods of evaluating the cytotoxicity of the nanoparticles, such as proliferation tests, apoptosis assays, necrosis assays, oxidative stress assays, and DNA damage assays, are described briefly.

### 6.1. Proliferation Assays

Proliferation assays are employed to analyze metabolically active cells in order to characterize cellular metabolism under exposure to NPs. The most often utilized tetrazolium salt for the in vitro toxicity assessment of nanoparticles is 3-(4,5-Dimethyl-thiazol-2-yl)-2,5-diphenyltetrazolium bromide (MTT) [52,53]. The procedure is advantageous since it produces quick results, can be performed in a multiplex format, and requires minimal modification of the model cells [54]. The assay relies on the detection of tetrazolium salt, which can be influenced by changes in culture media additions, media pH, ascorbate, and cholesterol levels. Because the MTT test also produces formazan, soluble dye-producing assays such as XTT or WST-1 are preferable [55,56]. Additionally, Neutral Red Dye, Resazurin and NRU assays are more likely to follow the metabolic activity mechanism. The major concern of these assays is their sensitivity and inability to detect viable cell numbers. Furthermore, some nanoparticles may adsorb in the same wavelength region as the dye used in the assay, which impedes the analysis. The concern for proliferation assays is that there is a need for a radioactive compound and harsh treatments of tissue sections [57]. For instance, the incorporation of [3H] thymidine is a method for assessing cellular proliferation, although it is avoided due to its toxicity and expensive cost.

### 6.2. Membrane Integrity Damage

The integrity of the membrane is utilized to determine the viability of the cells and the necrosis caused by nanomaterials. The absorption of a dye such as Trypan Blue can be used to determine membrane integrity. Trypan blue is a dye that enters dead cells but not living cells. Cell membrane stability can therefore be assessed using a Trypan Blue exclusion experiment [58,59,60].

Another approach consists of measuring cell leakage. For instance, cytotoxicity can be assessed by measuring the activity of cytoplasmic enzymes released by cells with damaged membranes. One such enzyme is the cytoplasmic enzyme lactate dehydrogenase (LDH), which is found in all cells. When the plasma membrane is damaged, LDH is rapidly released into the cell culture supernatant. The activity of LDH can be easily quantified by measuring the production of NADH during the conversion of the substrate lactate to pyruvate. During this reaction, the solution turns from yellow to red. The absorbance at 492 nm is directly proportional to the amount of LDH in the culture, i.e., to the number of dead or damaged cells [61]. In addition, ultrastructural observation using electron microscopy enables the detection of membrane damage and leakages of cell material [53].

### 6.3. Apoptosis Assays

In the in vitro assessment of nanoparticle toxicity, apoptosis is one of the most used indicators. Apoptosis and DNA damage are thought to be caused by excessive free radical production [62]. For instance, silver nanoparticles triggered apoptosis in mouse embryonic stem cells in vitro [63]. Apoptosis can be measured using the Annexin-V assay, Comet assay, TUNEL assay, and the inspection of morphological alterations [64]. Cell death indicators such as annexin-V and propidium iodide (PI) are commonly utilized in toxicity testing. The assay is based on Annexin-V’s affinity to bind to phosphatidylserine in the cell membrane. When bound to the membrane, Annexin-V fluoresces more brightly, indicating plasma membrane externalization [65]. In HeLa cell lines exposed to gold nanoparticles, apoptosis was induced using the Annexin V/PI [66]. The activation of the caspase-dependent process causes the plasma membrane to externalize. PI is an impenetrable dye that only stains the nucleus when the cell membrane integrity is compromised, which is associated with the late stages of apoptosis [67]. One of the most extensively employed methods for the detection of DNA damage in situ is TUNEL staining, which is performed in the TUNEL assay. It can be used to detect DNA damage associated with non-apoptotic processes, such as necrotic cell death brought on by hazardous chemicals. Nonetheless, the creation of false-positive results in detecting necrotic cells and cells undergoing DNA repair and gene transcription is always a challenge [68,69,70].

### 6.4. Oxidative Stress Assays

Nanoparticles may generate highly toxic reactive oxygen (ROS) and nitrogen species (NO) [71]. Different types of ROS may be generated under the interaction of NPs with water molecules. For instance, metal oxide nanoparticles were shown to produce singlet oxygen, superoxide radicals, hydroxyl radicals, and hydrogen peroxide by reducing oxygen dissolved in water. Interestingly, one type of NP can generate only one type of ROS, as in the case of MgO and CaO particles, which produce only singlet oxygen, or may produce a few different ROS, as is the case of ZnO NPs, which generate both hydroxyl radicals and hydrogen peroxide, while nanoparticles of CuO, ZnMgO and FeMnO_3_ can produce all types of ROS [56]. The reaction of 2,2,6,6-tetramethylpiperidine (TEMP) with O_2_, a stable radical that can be detected by X-band electron paramagnetic resonance (EPR), can be used to detect these free radicals. However, the method is expensive and demands sophisticated instrumentation [72]. The colorimetric C11-BIODIPY assay for lipid peroxidation and the TBA assay for malondialdehyde can also be used to assess oxidative stress. The availability of a variety of additional assays makes the evaluation considerably easier. These assays include the Amplex Red assay for measuring lipid hydroperoxide, the 5,5′-dithiobis-(2-nitrobenzoic acid) (DTNB) assay for measuring antioxidant depletion, and the Nitro blue tetrazolium assay for measuring superoxide dismutase (SOD) activity [73].

The production of peroxyl and hydroxyl radicals and nitrogen species can be quantified by the fluorescence of a 2′,7′-dichlorodihydrofluorescein diacetate (DCFH-DA) probe. DCFH-DA can easily traverse cell membranes and may be applied to detect ROS inside and outside of the cells. In contact with reactive species, DCFH-DA is hydrolyzed by a two-electron oxidation to the fluorescent DCF carboxylate anion, which can be monitored over time using a UV-vis spectrophotometer or using fluorescent microscopy [56].

## 7. Emerging Methods to Evaluate Toxicity of Nanoparticles

With the increasing quantity and variety of the new generation of synthesized nanomaterials, there is a great need for novel quick and reliable means to verify their safety [74]. Particularly, predictive models based on a large pool of reliable data are needed to establish intelligent testing procedures. Predictive methods such as high-throughput screening tests (HTS) and high content analysis (HCA) provide a risk analysis of nanomaterials by correlation grouping and read-across approaches with regulatory needs [75]. Some emerging methods developed for risk assessment, such as stem cell technology, tissue engineering, QSAR assays, molecular docking (MD), and MD simulation, are briefly described.

Various stem cell sources, such as fibromatosis-derived stem cells (FSCs), mesenchymal stem cells (MSCs), cardiac stem cells (CDCs), and embryonic stem cells, are currently available for toxicity assessment (ESCs) [76]. The successful integration of iPSCs with genetic diversity into cardiotoxicity in vitro testing has enabled the creation of the pluripotent stem cell-based model for evaluating the safety of engineered NPs [77]. HLCs derived from iPSCs have also been proposed as an alternative in vitro hepatotoxicity model for studying NP toxicity. The hepatotoxicity of silver NPs was assessed using HLCs generated from iPSCs [78]. Similarly, human iPSC-derived cardiomyocytes (hiPSC-CMs) were used to test the toxicity of ZnO NPs for cardiac safety [79]. Furthermore, iPSCs can be used as testing platforms for the toxic and therapeutic activity of NPs. In order to replace animal testing, optimal biopolymer-based 3D organ structures may be created and used to assess NP toxicity [80,81,82]. For cell growth and differentiation, 3D organ structure models involve a combination of synthetic or natural biological materials and stem cells that provide cell-to-cell interactions and suitable cell signaling pathways, as well as facilitate growth in all directions during the cell culture process [83]. Organ-on-a-chip systems fill the gap between conventional in vitro methods and animal and human studies. One of the main advantages of using organ-on-a-chip methods relies on their ability to produce high-fidelity models of human tissues and organs and their native microenvironment (such as the extracellular matrix, flow, geometry, and mechanical stiffness), which potentially provide new possibilities to systemically assess nanoparticle toxicity using established detection assays [84]. A recent study pointed out the toxic nature of cysteine-coated ZnO NPs using conventional cell culture experiments [85]. However, the dynamic conditions in the microfluidic lung-on-a-chip device indicated decreased cytotoxicity, suggesting the importance of considering organ-on-a-chip technology for assessing the toxicity of nanoparticles. Despite its advancing and diverse applications, organ-on-a-chip testing lacks standardization, which prevents comparisons between results from one study to the other.

Toxicity analyses using conventional cell-based or animal tests are fraught with ethical quandaries, together with financial and time constraints. As a result, computational toxicology is widely used in biomedical research to measure toxic effects on a variety of biological systems. In silico modeling is a fairly recent field that combines experimental and computational methodologies to provide a potent tool for deciphering systems at the atomic level. The method explores the physicochemical and structural characteristics of the nanomaterial as potential new pharmaceuticals for disease prevention and treatment. Currently, nano-specific databases, including the Online Chemical Modeling Environment (OCHEM), NanoDatabank, NanoHub, NanoMILE, and ModNanoTox, are available to conduct NP risk assessments [86]. In silico modeling based on bioinformatics and computational techniques can also be combined with experimental assays to provide a detailed method for measuring the potential risks of NPs. Molecular docking, quantitative structure–activity relationship (QSAR) studies, and molecular dynamics (MD) simulations are the three common types of computational approaches used in nanotoxicology studies as alternatives to animal testing [87,88].

Docking studies, in a nutshell, are simulation techniques that anticipate how small particles interact with large biomolecules such as proteins and nucleic acids. The first step in a docking study is to generate all possible conformations and orientations of each ligand based on the shape of the definite binding site in the protein structure. The second step entails using scoring functions to estimate favorable interactions between the protein and the docked ligand, leading to improvements. The docking scores calculated by the scoring functions are used after the docking study procedure to classify each correctly installed ligand in its binding site, which can then be used to define the highest affinity ligand for the target protein [89].

As an alternative to using animal models, some studies have recently used docking techniques to assess the binding conformations of ligands for toxicity evaluations [90]. The docking strategy can be used to investigate the chemical bonding of NPs with target enzymes. Molecular docking has also been used to assess the potential toxicity of various NPs with biological macromolecules such as CuO, TiO_2_, Fe_3_O_4_, Au, Ag, ZnO, Mn_2_O_3_, and Fe_3_O_4_ [89,90,91]. Although the application of molecular docking techniques to study the biological behavior of NPs is still relatively new, they have highlighted some common patterns of NP interactions with various metabolites and macromolecules [90]. As a result, docking interpretation between NPs and biological molecules has gained prominence in the field of nanotoxicology as an innovative method for predicting potential toxicity.

MD simulation is an in silico method that is widely used to investigate the chemical and physical properties of different molecular entities. This computational method enables the creation of toxicity prediction models that can orient NP design and development. As a result, this method provides an alternative strategy for investigating the toxicity of chemical compounds [92]. Modern MD simulations not only assist in understanding the time-dependent actions of atomic and molecular physical movements but also provide thermodynamic and kinetic properties of biomaterials at the nanoscale. MD simulations with all-atom resolution have been extensively performed to estimate the optimal levels of PEGylation in liposomes and other nanocarriers in order to optimize drug delivery efficiency [93,94]. Molecular dynamics modeling is a particularly powerful tool to assess the behavior of nanocarriers in the bloodstream, the efficiency of drug loading and controlled release, and the interaction of NPs with biological membranes and barriers [95]. Overall, MD simulations seem to be a perfect tool for obtaining molecular-level insights into precise nanocarrier interaction in a biological system [89]. However, the method demands supercomputing resources and thus, for many, is too expensive to be carried out.

Quantitative structure–activity relationship (QSAR) modeling is based on mathematical statistics and machine learning knowledge that allow for estimating the biological activity and toxicity of various chemicals. The QSAR model’s primary goal is to define an appropriate function that has a direct correlation between chemical structure and biological activity [96]. This has the potential to further summarize physiochemical and biological analyses to predict toxicity effects or establish ideal nanomaterials. Several QSAR modeling studies have been conducted to better comprehend the nanotoxicity of chemical substitutes such as NPs, metal oxides, and fullerenes [97,98,99]. To avoid animal testing, some governments use QSAR tools for toxic hazard prediction. In recent times, researchers have used QSAR models to assess the potential toxic effects of nanomaterial manufacturing [100]. Moreover, QSAR assays can be applied to predict the potential and efficiency of some drug delivery systems, as shown for heterolipids as delivery materials for nucleic acid therapeutics [101].

## 8. Challenges of New Approaches

The main challenges associated with nanoparticle-based emulsions are the sufficient protection of the integrity of the active drug, effective transportation across the biological barriers, and maintaining low toxicity. The toxicity of formulations has to be challenged not only on the cellular level but also in the context of affected tissues and under conditions of long-term exposure [102]. The advancement of in vitro and in silico tests has entailed technical challenges, such as establishing reliable culture conditions with in vivo-like levels to perform hazard testing. Non-mammalian alternative systems have emerged as an ideal strategy for overcoming the ethical concerns associated with traditional animal models in NP safety assessment. Non-mammalian models such as *Caenorhabditis elegans* (*C. elegans*), Drosophila (*Drosophila melanogaster*), the African clawed frog (*Xenopus laevis*), insect (*Galleria mellonella*), chicken chorioallantoic membrane (*Gallus gallus*), and zebrafish (*Danio rerio*) could be found to be acceptable approaches to guarantee the efficiency of toxicity assessments [103,104,105,106]. Nanoparticle-based drug delivery systems may suffer from limitations in terms of their low encapsulation efficiency, leakage before reaching the target, poor stability, and weak biological performance. Most published works have focused on the optimization of encapsulation methods and the improvement of the storage and thermal stability of the formulation. However, there are still methodological challenges in elucidating both in vitro and in vivo release mechanisms. Extensive research on new types of nanocarriers has shown that specific materials may improve the release performance. For instance, the self-assembled Amph-PVP nanoparticle was shown to successfully entrap indomethacin and deliver it to inflammatory sites, allowing for its prolonged release [107]. Similarly, 5-fluorouracil-loaded nanoparticles have shown promise as a delivery system in anticancer therapy [108], while N-vinylpyrrolidone polymer nanoparticles have been shown to be a promising drug delivery system since they are safe to use on both basal and activated endothelium [109].

## 9. Conclusions and Perspectives

Nanotechnology is a fast-developing science that entails the invention and development of nanoscale materials and devices. The use of nanotechnology in medicine enables us to tackle the problems and limitations of both drug delivery and diagnostics. The incorporation of an active pharmaceutical drug into a nanocarrier may prevent drug side effects and even increase the efficacy of conventional medications. Targeted nanoparticle delivery is now being researched extensively in cancer, inflammation, and infection treatments. Anticancer applications have been created for more than 20% of the therapeutic nanoparticles already in clinics or under clinical investigation. Moreover, the related research has focused on nanoparticle-mediated therapy for a variety of disorders, including neurodegenerative, infectious, autoimmune, and other diseases. Since 2009, the Food and Drug Administration (FDA) and the European Medicines Agency (EMA) have approved nano-drug formulations as therapeutic nanoparticle applications for targeted delivery systems in a variety of disorders [110]. It is critical to first analyze nano-based formulations in order to overcome these challenges in medicine. There are a variety of traditional procedures for evaluating these formulations, but they all have significant drawbacks, leading to the development of novel approaches for delivering medications more efficiently by evaluating the safety levels of nanoformulations. The toxicity is multifactorial as it depends not only on the formulation’s physicochemical properties and composition but also on the route of administration and dose. The increased use of nano-formulations necessitates greater attention to biological evaluations in order to ensure safety standards. Furthermore, a higher number of novel in vitro and in silica models should be supported in order to reduce the use of in vivo procedures to a minimum.

Many drug delivery systems have low efficacy, and usually, less than 5% of the injected dose is able to reach the targeted site [111]. This may arise from the structural heterogeneity of the targets and the biological barriers that limit the accessibility of the target. Nanocarriers sensitive to exogenous or endogenous stimuli (such as pH, temperature and redox potential) are therefore designed as an alternative to targeted drug delivery. The large variety of stimuli, together with a diversity of responsive nanomaterials, can be used to trigger drug release at the right place and time. Nevertheless, most targeted drug delivery and stimuli-responsive systems have limited chances of reaching the clinic stage because of insufficient biocompatibility and tissue accumulation caused by low degradability. In lieu of this, research at a comprehensive and collaborative level is important for the development of safe and efficient nanoparticle-based drug delivery systems in medicine.

## Figures and Tables

**Figure 1 pharmaceutics-15-00612-f001:**
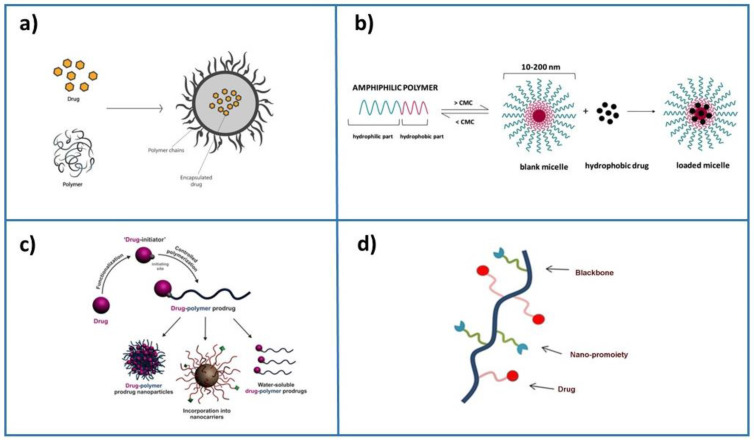
Principal formulation methods for drug encapsulation: (**a**) drug physical encapsulation (adapted with permission from [16]), (**b**) self-assembly of polymers and drug (adapted from [17]), (**c**) drug-initiated method (adapted from [12]), and (**d**) drug conjugation with a nano-promoiety.

**Figure 2 pharmaceutics-15-00612-f002:**
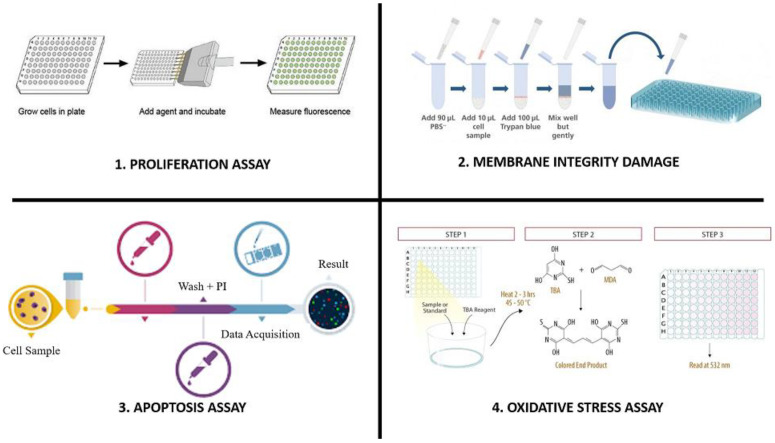
Conventional methods of evaluating the cytotoxicity of the nanoparticles.

**Table 1 pharmaceutics-15-00612-t001:** Conventional methods to demonstrate concerns for the bio-evaluation of nano-based formulations.

Nanotoxicity Evaluation	Conventional Methods	Mechanism	Concerns
Cytotoxicity	MTT Resazurin NRU assay Neutral Red Dye Thymidine Bromodeoxyuridine assays	Metabolic activity and proliferation assays	Detecting viable cell numbers is insufficiently sensitive, and dye interaction with NPs is a problem.
	TUNEL annexin-V Caspase assays	Apoptosis	False positives in recognizing necrotic cells and cells undergoing DNA repair or gene transcription.
	Trypan blue Propidium iodide Adverse outcome staining assays	Membrane integrity damage	These low-sensitivity techniques cannot be used.
Immuno-toxicity	ELISA RT-PCR	Antibody-antigen binding	Labor-intensive and expensive. Insufficient level of sensitivity.
Oxidative stress	C11-BIODIPY assay TBA assays		Indirect methods.

## Data Availability

Not applicable.

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
