# Peer review of "Experimental Methods for the Biological Evaluation of Nanoparticle-Based Drug Delivery Risks"

_pharmaceutics, 2023, doi:10.3390/pharmaceutics15020612_

Round 1

Reviewer 1 Report (Previous Reviewer 1)

The paper does not follow through adequately on what it promised to do. Still, there is a potential value in what the manuscript proposes nevertheless the authors fail to add novelthy. Recent advances in nanoparticles biocompatibility studies are missed.

Author Response

In the introduction, we promised to provide a perception regarding the risks associated with nanoparticle-based formulations, with a particular focus on the assays applied to assess the cytotoxicity of nanomaterials. We provided a global overview of nanoparticle applications, barriers, and toxicity, as well as a biological evaluation of nanoparticle-based formulations, for this. We would like to thank you for your comment that some recent advances in nanoparticle biocompatibility were missed. We improved the revised version by adding additional references. We specifically stated that changing the chemical composition of nanomaterials can improve biocompatibility, as seen with TiO2 NPs doped with Fe-ions. Rizzotto et al.2022 demonstrated that mixed metal oxide Fe2TiO5 NPs are extremely cell friendly. In addition, coating a nanomaterial surface with organic molecules such as proteins, fibers, and carbohydrates significantly improves their biocompatibility. This feature was observed with nanoparticles obtained by green synthesis (added new references in the contents of 4.4).

Reviewer 2 Report (Previous Reviewer 2)

It is clear that the authors of "Biological Evaluation of Nanoparticles-Based Drug Delivery Risks" have put consideration time into the revision of this manuscript. While there is nothing technically wrong with this manuscript and could be published, the two main concerns that I have about this article are:

1) there seems to be a very broad range of topics that are included, all of which include minimal details. The authors may want to consider focusing in on a particular subsection and providing more details (some sections are only 3 sentences long). 

2) can the authors explain how this review is unique from other reviews that have been published in this area?

Author Response

Response 1.: We would like to thank you for this remark. We compiled the short subheadings 6.2 and 6.4 in the revised version of the manuscript.

Response 2: Nanoparticle-based formulations have successfully entered the clinic for the delivery of different drugs and vaccines. This results from the extensive work in the development of such technologies, which is summarized in many review articles. Our review is unique in that it focuses on the risk aspects of nanoparticle-based formulations. Generally, review papers deal with one type of nanoparticle or with different nanoparticles developed to treat a specific disease, such as cancer, Alzheimer's disorders, antimicrobial drugs, etc. We provide an overview of different nanoparticle-based formulations without any restriction on specific disease treatment. Another unique feature of the overview we provide is the presentation of current risk assessment technologies alongside emerging and still-under-development methods. 

Reviewer 3 Report (Previous Reviewer 3)

Authors have made a sincere effort to revise the manuscript in light of earlier comments, and the manuscript has improved. I would recommend its publication, but the manuscript needs to be edited carefully for flow and consistency. The authors should seriously consider getting help from the Editorial office.

Author Response

Response: Thank you so much for your encouraging comments. We will take help from the editorial office for the flow and consistency of the manuscript.

This manuscript is a resubmission of an earlier submission. The following is a list of the peer review reports and author responses from that submission.

Round 1

Reviewer 1 Report

The novelty of this review is overall poor. Please try to better outline the novelty of the investigations  here reported providing a broader and deeper discussion of the recent advances in the field. The manuscript would need a thorough linguistic revision.

In this form the paper, however, does not follow through adequately on what it promised to do. Still, there is a potential value in what the manuscript here proposes and I encourage the authors to flesh out the paper’s ideas more thoroughly. 

Reviewer 2 Report

The main concern noted during review was that this review paper does not significantly add to the current body of knowledge available. Other concerns noted during review were 1) no consistent theme (topics jumped all over the place), 2) Section 3 contains no references, 3) the figures did not add any value and should be removed and redone to be meaningful, and 4) the sections on methods are unnecessary as they add no new information 

Reviewer 3 Report

The topic for this review is interesting and it could serve a good purpose if the authors had made it an authoritative review of this area. Nanoparticles (NPs) are playing a key role in biology, and there are numerous review articles that are written on the topic. Each makes a contribution in its own right. This manuscript provides general long statements on varied aspects of biological evaluations, but not a focused overview of any aspect: it talks about toxicity, but does not go into detail here; biological barriers are just touched upon with no in-depth analysis of what has been achieved and what more should be expected as high priority; physical characteristics of NPs are key in the expected outcomes, but these are just listed in few sentences with no examples, details etc.; Fig 3 has no purpose but just takes space; varied essays are described but with the view of more practicality than what comparative advantages each offers. Conclusions list again only general features and not what is the state-of-the-art and how it could be improved upon.

Overall, I feel that the review is too wordy with very few visual representations and lacks a detailed expert view of the field. Authors should make the manuscript more appealing for the reader, as their efforts will be rewarded by a good contribution to the area.